# Employees’ Perception of HRM Practices and Organizational Citizenship Behaviour: The Mediating Role of the Work–Family Interface

**DOI:** 10.3390/bs12090301

**Published:** 2022-08-24

**Authors:** Maria Luisa Giancaspro, Silvia De Simone, Amelia Manuti

**Affiliations:** 1Department of Education, Psychology, Communication, University of Bari, Via Crisanzio 42, 70121 Bari, Italy; 2Department of Pedagogy, Psychology, Philosophy, University of Cagliari, Via Is Mirrionis 1, 09123 Cagliari, Italy

**Keywords:** human resource management strategies, work–family interface, organizational citizenship behaviors

## Abstract

The aim of the study was to explore if and to what extent a positive perception of Human Resource Management Practices could be related to Organizational Citizenship Behaviors and if the Work–Family Interface could act as a mediator of such relationship. A group of 406 employees of an Italian sector of the Public Administration filled in a self-report questionnaire encompassing socio-professional information and the following psycho-social measures: Perception of Human Resource Management Scale, Work–Family Interface, and Organizational Citizenship Behaviors Scale. The results confirmed the importance of Human Resource Management Practices perceptions for positive Organizational Behaviors underlining the crucial role played by positive work-to-family and family-to-work spillover as mediators. The study reflected on the work–family relationships demonstrating the mediating role of positive work–family spillover in the relationships between perceived HR practices and organizational citizenship behaviors. The main limitations were related to the cross-sectional nature of the study and to the self-report measures used that raised concerns about possible common method bias. The findings showed implications for HR practices to plan actions and interventions aimed at supporting employees’ work–family balance and at encouraging organizational citizenship behaviors.

## 1. Introduction

Within the last decades, work–life balance programs have become one of the most significant Human Resource Management (HRM) practices, both for their impact on employees’ wellbeing, commitment, and motivation, and for their implications in terms of organizational performance and economical success. This evidence was largely supported by theoretical developments and good managerial practices showing that ‘happy workers are productive workers and productive workers are likely to be happy’ [1], thus further encouraging Human Resource Management practitioners operating in private and public contexts to invest in these kinds of people-oriented interventions.

Basing on these assumptions, evidence shows that the need to balance work and family engagements heavily influences employee attitudes and behaviors [2]. Consistently, organizations have started investing in the development of focused interventions to help employees managing demands coming from the family and from the working contexts. Good examples of such practices are the introduction of flexible working arrangements, job design interventions or smart working, that were showed to have positive feedback on employees’ positive organizational behaviors [3,4].

Accordingly, further empirical contributions revealed that social support perception [5], work-life benefits and policies, a family-friendly organizational culture [6], and formal and informal practices of work–life balance [7] concurred to reduce individual perceptions about work–family conflict, thus creating a supportive work environment and improving performance. 

Among the individual and organizational outcomes that positively related to work–life balance interventions, job satisfaction, commitment, engagement, and citizenship behaviours (OCB) were those that received more attention [8,9]. Therefore, this vivid scientific debate confirmed the crucial role played by HRM in considering employees’ needs in terms of balance between their multiple engagements in work and family, sometimes experienced as conflicting identity dimensions and sometimes as mutually enriching domains.

This two-fold nature of the relationship between work and family was reflected in the literature that attempted at theorizing work–family relationships according to many different research perspectives over time (see for instance the meta-analysis review [10,11,12]). 

This still open debate led to the conceptualization of work–life balance, namely the degree to which an individual can simultaneously balance the emotional, behavioral and time demands of both paid work, family and personal duties [13], in contrast to work–life conflict that occurs when involvement in one domain, for example work or personal life, interferes with involvement in the other domain [14].

Studies in the field supported the progressive theorization of work–family conflict as a composite construct comprising two distinct but related dimensions, that underlined the direction of the perceived interference: work interference with family and family interference with work [15]. These two dimensions were related to different antecedents and consequences. Antecedents of work–life conflict were categorized into three main groups [12,16]: work domain variables (e.g., organizational support, conflict, stress at work, work engagement, job satisfaction, commitment), non-work domain variables (e.g., marital conflict, family support, number of hours spent on housework or childcare, having children at home and age of youngest child), and demographic and individual variables (gender, personality, income). Similarly, consequences of work–family conflict were distinguished into three different categories: work-related, family-related, and domain-unspecific outcomes [11]. Several meta-analyses showed the strict link between perceived work–family conflict, and specific work attitudes such as job satisfaction, e.g., [17], commitment, e.g., [18], engagement [19], and organizational citizen behavior, e.g., [20]. Traditionally, studies on the relationship between work and family focused more on the negative aspects of work–family conflict [21], whereas the positive effects were only recently explored [22]. Despite moving from the stress perspective [23], most studies, in recent years, emphasized the positive exchange that could be established between work and family domains and the potential enrichment that could derive from this exchange for both individuals and organizations [24]. This alternative and positive reading of the relationship between work and family also showed many significant effects on job attitudes, on job performance, and on job satisfaction [25]. Yet, these studies considered the participation in multiple roles as an opportunity to expand experiences and resources [23].

Consequently, scholars’ attention shifted from a perspective cantered only on the negative, conflicting, and stressful aspects of the relationship between the family and work domain to a positive “spillover” perspective where moods, skills, values, and behaviors were thought to be potentially transferable from one domain to another [22].This perspective linked to what was called the spillover model shed light on the enrichment that might derive from the participation to both contexts in terms of resources that could be exchanged to adjust to different challenging demands widening experience and improving wellbeing [26].

In line with these speculations, work and family demands were also re-read according to an instrumental model, postulating that activities in one sphere could be also beneficial to the other and consequently fostering the formulation of the so-called compensation model, considering work and life as domains that could balance each other so that what may be experienced as lacking in one sphere, in terms of demands and/or satisfactions, could be made up in the other (see [27] for an overview). The same assumption is at the base of the work–family enrichment approach considering “the extent to which experience in one role improves the quality of life namely performance or affect, in the other role” [26] (p. 6).

More recently, Clark [28] proposed the border model maintaining that “people are daily border-crossers as they move between work and home” (p. 259), and posed interesting questions about the permeability of borders and the subjective ability to manage them.

Following this perspective, some authors [29,30,31] integrated the positive and negative aspects related to work and family demands, proposing a four-factor model that took into account the complexity of this relationship [32,33].

Accordingly, very little is known about the relationship between HRM practices perception, work-to-family interface, and employees’ positive behaviors. Within such a framework, the study aimed to fill a gap in the literature and to examine the role of work–family interface dimensions in the relationships between Perception of HRM and Organizational Citizenship Behaviors.

In this vein, the present study was intended to fulfil two aims: (1) to give a contribution to theory development investigating the relationship between employees’ perception of HRM practices, work–family interface, and organizational citizenship behaviors, (2) to provide empirical evidences about the relationship among these variables, specifically examining the mediating role of work–family in the relationship between HRM practices and organizational behaviors. 

According to these aims, the paper was structured as follows. The first section described the conceptual framework of the research, while the second one discussed research design and results coming from a study conducted with an Italian group of employees working in the Public Sector context. The final section of the paper was devoted to drawing the implications of the study for theory and practice in the field of human resource management and development, providing managers and organizational decision makers with insights about concrete people management actions and welfare interventions.

## 2. Conceptual Framework and Research Hypotheses Development

### 2.1. Human Resource Management Practices and Employees Outcomes

Some recent developments in the field of Human Resource Management showed that employees’ perceptions about people management strategies played a crucial role in determining positive attitudes and behaviors [34,35].

According to the exchange theory [36], relationships between employers and employees are mostly based on social and economic exchanges [37]. Economic exchanges are specified by contractual arrangements and define aspects such as pay, working hours, and job entitlements [38,39], whereas social exchanges refer to the development of the interdependent relationship between persons and organizations [40], basically regulated by “normative rules” of reciprocity [41]. In view of the above, successful social exchange relationships are those characterized by high degrees of mutual loyalty and trust [42].

Social exchange theory has been used extensively as a framework to explain the relationship between what are labelled as High-Performance Human Resource Practices (HPHRP) and employee outcomes. High-performance Human Resource Practices (HPHRPs) are typically a group of coherent, interrelated HR practices designed to promote employee motivation and commitment [37,43]. HPHRPs are conceived as strategically planned combinations of HR practices meant to improve performance [44,45,46,47,48]. Some authors also refer to them in terms of high commitment practices [49,50,51,52,53], underlining the scope of such work systems designed to foster employee commitment, control/efficiency, or involvement. This would happen because they are social exchanges involving the development of interdependent relationships in which unspecified bidirectional transactions occur. In other words, “something” desirable is given by the “donor,” and at some future point in time, “something” desirable is returned by the “recipient” [38]. Such interdependence is based on “normative rules” of reciprocity [36], which are the “defining characteristics” of social exchange relationships [41] (p. 876). Because of the temporal gap between what is given and what is returned, successful social exchange relationships are characterized by high degrees of loyalty and trust between donor and recipient [38]. 

The implementation of these high-performance or high-commitment Human Resource practices is linked to the so-called resource-based or soft approach to HRM, arguing that to invest in human resources could make a return in terms of commitment which is intangible but strictly related to performance and competitiveness. This view is opposed by the control-based or hard approach which conversely postulates the need to constantly monitor and direct employees.

Within the last decades, abundant empirical evidence showed that when organizations invest in HPHRPs, employees perceived this effort as an expression of the organization’s trust and commitment to them, as an appreciation of their work, and as a desire to engage in a long-term relationship [54] rather than in a short-term economic exchange relationship with employees [37]. Most perceived HRM studies have showed job satisfaction, affective organizational commitment [55], and work engagement [53,56] as some of the main attitudinal outcomes of the person/organization relationship, such as organizational citizenship behavior (OCB) [57,58] and job performance [56]. OCB has been defined as “individual behavior that is discretionary, not directly or explicitly recognized by the formal reward system, and that in the aggregate promotes the effective functioning of the organization” [59] (p. 4). Due to this extra role and discretionary nature in the definition, the subsequent studies mostly treated OCB as constructive, self-initiated, spontaneous, or voluntary behavior aimed at enhancing the productivity of the workplace [60]. Organ’s [59] conceptualization of OCB includes five behavior types—altruism, courtesy, sportsmanship, conscientiousness, and civic virtue—all necessary for effective organizations and enhancing effectiveness of the organizations. These five dimensions cover such organizational behaviours as helping co-workers, following company rules, not complaining, and actively participating in organizational affairs. Altruism means helping other members of the organization in their tasks; for example, voluntarily helping new employees, helping co-workers who are overloaded, assisting workers who were absent, guiding employees to accomplish difficult tasks. Conscientiousness is a discretionary behaviour that goes well beyond the minimum role requirement level of the organization, such as obeying rules and regulations, not taking extra breaks, working extra-long days [61]. Sportsmanship is defined as “a willingness to tolerate the inevitable inconveniences and impositions of work without complaining” [62] (p. 96). It refers to person’s desire not to complain when experiencing the inevitable inconveniences and abuse generated in exercising a professional activity. Courtesy refers to the gestures that help others to prevent interpersonal problems from occurring, such as giving prior notice of the work schedule to someone who is in need, or consulting others before taking any actions that would affect them [62]. Civic virtue refers to the constructive involvement in the political process of the organization and contribution to this process by freely and frankly expressing opinions, attending meetings, discussing with colleagues the issues concerning the organization, and reading organizational communications such as mails for the wellbeing of the organization.

Yet, many studies considered organizational citizenship behaviors as a positive outcome of human resources management practices perceptions in different organizational contexts [34,37,63,64,65,66]. For example, even though recent work by Shen and Benson [67] established that SR-HRM positively influences employee OCB through eliciting higher levels of organizational identification. For instance, Pham, Tučková, and Chabbour [68] conducted a study in the hospitality industry showing the positive relationship between employees’ perception of green human resource management, defined as “human resource management activities”, enhancing positive environmental outcomes [69] (p. 1075) and refers to the human resource management aspects of environmental management [70], and their OCBs for the environment. In a similar vein, Watty-Benjamin and Udechukwu [71] proved the same relationship with employees of the public sector. Furthermore, Husin, Chelladurai, and Musa in 2012 [72] showed that high-commitment HRM practices influenced OCBs which, in turn, were associated with Perceived Service Quality, that is considered a perceived attribute based on the experience of the customer regarding the service that the customer perceived during the delivery process of the service [73].

Basing on these assumptions, the present study assumed that: 

**H1.** 
*A positive perception of HRM Practices is related to organizational citizenship behaviors.*


Despite plenty of studies supporting the relationship between HRM practices perception and organizational citizenship behaviors, no evidence confirms a linear relationship between the two. Therefore, a variable that was proved to significantly mediate this relationship is the work–family interface [74,75].

### 2.2. Employees’ HRM Practices Perception, Work–Family Interface, and Organizational Citizenship Behaviors 

Some of the most recent studies examining the relationship between HRM and the work–family interface suggested a positive influence played by HRM practices perception on work and family spillover [76,77,78,79] Some of these studies also investigated the extent to which work and family interface might relate to organizational citizenship behaviors as employees’ positive outcome [78,80].

Batt and Vallour [77] explored the relationship between employees’ perception of Human Resource practices (e.g., work-family policies, HR incentives, work design, etc.) and different outcomes including work–family conflict and employees’ control over managing work and family demands. These studies showed that work design characteristics were associated with employees’ control perception over managing work and family demands, and that HR incentives were associated with the work–family conflict [81].

In a similar vein, Bakker and Geurt [76] highlighted that those employees enjoying opportunities for development, autonomy, and performance feedback tended to preferably consider the relationship between work and family through an enrichment perspective. More recently, results from an Italian study aimed at assessing the relationship between dependent and self-employed workers’ work–family enrichment and well-being showed that opportunities for professional development mediated the relationship between supervisor support, job security, and work–family enrichment [79].

Baral and Shivganesh Bhargava [82] emphasized that job characteristics and supervisor support were positively related to work-to-family enrichment and that work-to-family enrichment mediated the relationships between job characteristics and job outcomes and between supervisor support and affective commitment. These findings found that when workers perceive their jobs to be higher on organizational core dimensions (i.e., autonomy, variety), they perceive higher work-to-family enrichment, which lead to improved job satisfaction, affective commitment, and organizational citizenship behavior.

Further, Carlson and colleagues [78] found that when job security is high, individuals are not distracted by worry or exhausted by tension and tend to assume more promptly the responsibility both inside and outside the workplace, thus creating more opportunities for family–work enrichment.

In view of the above, the study maintained that:

**H2.** 
*A positive perception of HRM Practices is positively related to Positive Work-To-Family Spillover (H2a) and to Positive Family-To-Work Spillover (H2b).*


Abundant research focused on the relationship between work and family interactions and organizational citizenship behaviours, and both directions of work–family conflict were found to be associated with this positive behavioural outcome [20].

Thompson and Werner [83] showed that higher levels of role conflict in the workplace were related to lower levels of OCBs. More specifically, the authors showed the direct effects of work–family role conflicts on OCBs and that organizational commitment mediate the relationship between role conflict and the OCBs dimension of loyalty. In a study conducted with a group of teachers, Bragger, Rodriguez-Srednicki, Kutcher, Indovino, and Rosner [80] found that workers reporting higher levels of work–family conflict showed lower levels of organizational citizenship and vice versa. The study also highlighted how work–family culture positively predicted crucial employees’ outcomes such as organizational commitment and OCB.

Investigating a sample of 205 supervisor/subordinate dyads, Carlson, Kacmar, Grzywacz, Tepper, and Whitten [78] concluded that subordinate work–family balance predicted supervisors’ appraisals of subordinate’s engagement in both organization and individual OCB and that this relationship was fully mediated by the positive effect rated by the subordinate.

Similarly, interventions aimed at creating a family-friendly culture produce positive benefits on job satisfaction, commitment [82,84,85], and on organizational citizenship behaviours [80,86,87].

Based on the above, the study assumed that:

**H3.** 
*Positive Work-To-Family Spillover (H3a), and Positive Family-To-Work Spillover (H3a) are positively related to Organizational Citizenship Behaviours.*


In this perspective, the present study considered positive perceptions of HRM (related to training and development, safety, rewards and career development, growth opportunity, evaluation, and feedback) as resources that can reinforce positive work–family spillover. Therefore, the work–family interface was assumed to mediate the impact of HRM practices perception on organizational citizenship behaviours, as suggested by some empirical evidences, assessing the relationship between job demands, work–family interface, and employees’ OCBs [88] and more specifically the link between organizational interventions for work–life balance, (e.g., work–life benefits and policies, supervisor support, and work–family culture) and job outcomes, such as job satisfaction, affective commitment, and organizational citizenship behaviour [82]. Therefore, the present study finally assumed that:

**H4.** 
*Positive Work-To-Family Spillover and Family-To-Work Spillover would mediate the relationships between Perception of HRM and Organizational Citizenship Behaviours, as shown in Figure 1.*


## 3. Materials and Methods

### 3.1. Sample and Design

The research was conducted in a regional division of an Italian sector of Public Administration. All regional offices and the central management of the Italian Revenue Agency (in Italian “Agenzia delle Entrate”) were involved. It is a context in which although there is personnel management and formalized work–life balance policies, these policies often struggle to turn into HR practices also because of the significant presence of male workers in the company population and in management and top positions. For this reason, it was particularly interesting to carry out this study within this specific context. 

A sample of 406 employees participated to the study. 212 of them were males (52.2%), aged between under 30 and 51 and 60 years (43.3% is aged between 51 and 60), and with service seniority ranging from 6 to 25 years (6–15 years = 32.5%; 16–25 years = 31.3%).

Participants were contacted through the HR Department after the research team was authorized by the management. An email was sent through the intranet communication system to all employees, who received a link to fill in the questionnaire. A cover letter was also delivered to them describing the research aims and assuring confidentiality. The response rate was 80%. Before filling in the questionnaire, participants were invited to sign the informed consent according to the Italian data law on data protection (Legislative Decree No. 196/2003). The Ethical Committee of the University of Bari was informed about the study and gave its consent (ET-20-15).

### 3.2. Variables and Measures

Data were collected through a structured questionnaire composed of a socio-demographical section, encompassing information about gender, age, education, typology of contract, and seniority, and a second section, composed by the psycho-social measures used to assess the variables focused by the study.

Where available, we adopted the Italian validated versions of the measures considered. Conversely, when no Italian version was available, the original measures were translated following the back-translation technique. Two mother tongue researchers independently translated the scales and then compared their versions by calculating an agreement index [89]. 

The variables and measures investigated by the study were the following:

HRM perception. This variable was assessed through the 9-item scale by Gould-Williams and Davies [42] that measures employees’ perception about the human resource management practices implemented by the organization. Specifically, the scale measures perceptions about training and development initiatives (e.g., “The organization provides me with sufficient elements for training and development”), about job security (e.g., “I feel my job is safe”), about compensation (e.g., “The rewards I receive are directly related to my job performance”), about career management opportunities (e.g., “Career development is considered a top priority in this department”), and about feedback and immaterial rewards (e.g., “I receive significant feedback on my performance at least once a year”). Participants were invited to express their agreement/disagreement with each item using a 5-point scale, from “not at all” (1) to “completely” (5). 

Work–family interface. To assess this variable, the Italian version of the Work–family interface Scale (WFIS) validated by De Simone and colleagues [29] was adopted. The scale encompasses 14 items referred to in four dimensions; in this research, we only used two positive dimensions: positive work-to-family spillover (POSWIF) (e.g., “You come home cheerfully after a successful day at work, positively affecting the atmosphere at home?”) and the fourth considers positive family-to-work spillover (POSFIW) (e.g., “After spending time with your spouse/family, you go to work in a good mood, positively affecting the atmosphere at work?”). Participants were invited to indicate how often did they experienced each of the described situation using a 5-point scale from 1 (never) to 5 (very often).

Organizational Citizenship Behaviors. The Italian version of the Organizational Citizenship Behavior Scale originally elaborated by Podsakoff, MacKenzie, Moorman, and Fetter [90], and validated by Argentero, Cortese, and Ferretti [91], was used to measure this variable. The scale is composed of 24 items (e.g., “Help others who have heavy workloads”) measuring the occurrence of extra-role discretionary behaviors. Responses were assessed through a 5-point scale, ranging from “never” (1) to “always” (5). 

## 4. Results

### 4.1. Preliminary and Descriptive Analysis 

To test the role of work-to-family and family-to-work spillover in the relationship between Perception of HRM Practices and the Organizational Citizenship Behaviour, a parallel multiple mediator model was applied [92], using the PROCESS SPSS (Model 6) computational tool [93]. Given the limited sample size and to prevent violation of normal distribution assumptions, the non-parametric bootstrapping method was used as a robust estimation of both direct and indirect effects [92]. Bootstrapping provided a confidence interval (CI) around the indirect effect of the independent variable (HRM perception) on the dependent variable (Organizational Citizenship Behaviour) via the mediators (Positive work-to-family spillover and negative work-to-family spillover). Multiple mediations are significant if the interval between the upper limit (UL) and lower limit (LL) of a bootstrapped 95% CI do not contain zero, which means that the mediating effect is different from zero [94]. 

Because a single questionnaire was used to collect data through self-report scales, common method variance was addressed following some recent indications [95,96], specifically suggesting how to protect item consistency, social desirability, and to reduce evaluation apprehension. Among all statistic methods, Harman’s single-factor test was used in this case. From the result, the Total Variance Explained of the first component accounts for less than 50% of all variables in the model and so the instrument is free from significant common method bias effects. Moreover, items were inserted randomly into the questionnaire and scales were graphically separated from each other.

Before investigating the hypotheses of the study, some preliminary analyses were conducted. Table 1 shows the distribution of mean scores and standard deviations for each variable and Pearson correlations between the constructs that were chosen to assess positive and negative spillover, HRM practices perception, and organizational citizenship behaviours. The correlation analysis (see Table 1) showed significant bivariate relationships between HRM practices, OCB, and the two positive dimensions of work–family spillover. On the other hand, HRM practices showed a significant correlation with the two positive dimensions of work–family spillover [97]. Therefore, this study explored the mediating role of the positive family-to-work (POSFIW) and positive work-to-family (POSWIF) spillover in the relationship between HRM practices and OCB.

### 4.2. Mediation Analysis

A model with parallel mediators was tested to investigate the mediation role of POSWIF and POSFIW in the relationship between HRM practices and OCB, controlling for the effects of gender and age. The results of the mediation model (see Figure 2) showed significant direct effects of HRM practices on POSWIF and POSFIW. Furthermore, the two positive dimensions of work–family spillover showed significant effects on OCB. As regards to the relationship between HRM practices and OCB, the model showed that the direct effect was not significant, but the relationship was totally mediated by POSWIF, β = 0.03, 95% CI. (0.01, 0.04), and POSFIW, β = 0.06, 95% CI. (0.01, 0.07). As regards to control variables, age did not show any significant effect whereas gender showed a significant effect only on POSWIF, suggesting that women may have higher levels of positive work-to-family spillover. This model explained 9% of the variance of OCB, 8% of the variance of POSWIF, and 15% of the variance of POSFIW.

The results from the linear hierarchical regressions showed a total confirmation of the hypotheses. More in detail, as regards to H1, HRM practices perception was proved to be a significant predictor of OCB (β = 0.176; *p* = 0.000). Likewise, HRM practices perception showed a positive relation with positive work-to-family spillover (β = 0.249; *p* = 0.000) and with positive family-to-work spillover (β = 0.466; *p* = 0.000). Further, positive work-to-family interface (β = 0.111; *p* = 0.000) and the positive family-to-work interface (β = 0.137; *p* = 0.000) were positively related to OCB. Therefore, H2a and H2b were both confirmed. 

Consistent with H3 hypotheses, HRM perception showed significant and positive direct paths to positive work-to-family and positive family-to-work spillover, as displayed in Figure 2. Positive work-to-family and positive family-to-work spillover showed also significant paths to OCB. Moreover, in the mediated model, HRM perception lost the direct and positive effect on OCB. At the same time, it had significant indirect effects on OCB as mediated through positive work-to-family spillover (β = 0.018, CI = 0.006, 0.036), for the 40% of the effect and positive family-to-work spillover (β = 0.031, CI = 0.009, 0.060), for the 46% of the effect. These results show that positive work-to-family and family-to-work spillover fully mediated the relationship between HRM perception and OCB. 

In view of the above, results coming from the study confirmed the crucial role played by HRM practices perception in influencing organizational citizenship behaviour. In this vein, the study showed that supportive and employee-centred HRM practices could contribute to strengthening workers’ motivation, encouraging them to perform better and more efficiently whilst also engaging in extra-role behaviours. Likewise, positive work-to-family and family-to-work interfaces were proved to be important mediating factors contributing to enhance the Person/Organization fit. Accordingly, employees perceiving support, consideration, and attention from their organization and who succeed in positively exploiting the contamination between family and work and vice versa, are actively encouraged to also perform organizational citizenship behaviours.

## 5. Discussion

Within the current global market scenario, human factor is increasingly being considered a significant and crucial factor for every kind of organization, being it a commercial, educational, or governmental, public, or private organization [98]. Accordingly, the viability of the organization, the achievement of its goals, as well as the motivation, well-being, and performance of its employees, strictly depend on the efficacy of its human resource management system [99]. Consistently, abundant literature on the so-called ‘High-Performance Human Resource Practices’ contributed to showing how people based HRM practices, addressed to allow employees to express themselves through their work, participating in organizational processes and concurring to decision making would result in higher engagement, satisfaction, and in better performance [37,100,101]. HRM practices can set the quality of employer–employee relationships, thereby motivating employees to exhibit organizational citizenship behaviors and encouraging their personal contribution. This evidence is true for the private sector, on which most research developments focused on, but also for the public sector, being public administration a crucial context of vital importance for the development of a country, as testified by extant research [102,103].

In view of the above, the present study aimed to contribute to this debate. The results confirmed H1 that suggest the significant relationship between HRM practices perception and OCB, as shown by prior research in the field. This means that since HRM practices have the potential to guide and even define the nature of the relationship between employee and employer, they could thereby also serve as a source of motivation for employees to exercise OCB [104,105,106,107]. Employees’ involvement in the decision-making process enhances their sense of importance [108]. Exposing employees to intensive training and paying attention to their career path planning all reinforce the message that the organization is committed to their individual development and growth. Incentives and rewards based on performance evaluation enhance workers’ sense of recognition and fair treatment, which, in turn, increases their loyalty to the institution [109,110]. HRM practices thus have the potential to offer certain inducements to employees, who in return are more likely to exhibit OCB [111].

The results also confirm H2a, the positive effect of HR practices on the positive work–family spillover, and H2b, the positive effect of HR practices on the positive family–work spillover. This data indicates that a positive work environment in which there is attention to people, their growth, and their development promotes not only employees’ well-being, but also the transfer of positively valanced affect, skills, behaviors, and values from the work domain to the family domain, thus having beneficial effects on the family domain [14]. After all, family-friendly work environments promote positive outcomes for employees, energizing them at work and increasing positive mood at home [112,113,114]. Previous research showed that organizations, through the provision of job resources (e.g., job autonomy, sense of control, co-worker, and general supervisor support) can help employees to realize a positive spillover between work and family [25,115,116,117]. More generally, family-supportive supervisor behaviors [118,119] can create work-to-family positive spillover effects, largely through supervisors’ ability to stimulate employee work engagement. In the same way, it seems that if employees perceive themselves to be placed in a supportive and comfortable context in which human resource management practices are people-oriented, they will feel more inclined to share positive feelings associated with family domain and life satisfaction also in the workplace.

Furthermore, the relationship between the work–family interface and organizational citizenship behavior (OCB) also confirmed (H3). Specifically, it seems that both the positive work–family and family–work spillover have a positive effect on organizational citizenship behaviors. If, as Yu, Wang, and Huang [120] suggest, the interference of work in the family and vice versa causes a reduction in resources and energies and therefore a decrease in OCB, in this case, the positive contamination in both directions generates an enhancement of resources with a greater aptitude to engage extra-role behaviors.

Moreover, the study provided empirical support to the mediating role played by positive work-to-family and family-to-work spillover paving the way for further developments of theory and HRM practices (H4). Yet, although most studies adopting the work–life conflict perspective confirmed that experiencing a difficulty in managing double roles and multiple engagements in work and life might lead to negative work behaviors such as stress, emotional exhaustion, and burnout [121,122], the results coming from the present study suggested that a positive work-to-family and family-to-work spillover might result in positive emotions that could potentially promote employees’ availability to perform extra-role behaviors, as confirmed also by some most recent contributions [123,124].

Accordingly, the study granted empirical support to a growing interest showed by managers and practitioners to develop welfare policies that would consider employees’ needs in terms of work–life balance as an important part of the social exchange relationship with their workforce [125,126]. In this direction, the study could give a scientific contribution to the development and implementation of HRM practices addressed to enhance positive spillover and contamination between life and work spheres and to encourage employees’ thriving at work [127].

This contribution is far more important if framed within the “new normal” scenario mostly featured by the challenges posed by remote working to work/life balance. Accordingly, although planned to be a beneficial HRM tool to support employees in balancing life spheres’ demands, a recent review of 40 empirical studies analysing work–life balance and working from home during the pandemic showed several misfits between desirable expectations and the undesirable realities of remote work: (1) flextime vs. work intensity, (2) flexplace vs. space limitation, (3) technologically-feasible work arrangement vs. technostress and isolation, and (4) family-friendly work arrangement vs. housework and care intensity [128]. 

Therefore, in line with this evidence, although fully encouraging, the results coming from the present study, which moreover was conducted in the Italian Public sector where remote working modalities are increasing after the pandemic, confirmed the need to highlight the crucial role Human Resource Development practitioners can play in assisting employees to achieve a fit between their expectations and experiences in traditional, remote and/or hybrid work settings, maximizing the advantages, balancing needs, and ensuring sustainability to one’s own careers [129,130,131]. 

Yet, this meaningful shift in “how we work” is not simply a matter of objective work modalities and conditions, rather it would entail implications for core HRD topics, including learning and development, performance management, workload, effective communications and relationships, and people management capability. In this vein, to be effective and to produce the virtuous relationship between Human Resource Management, employees’ wellbeing, work–life balance, and positive organizational behaviors that even the present study has shown, any HRM practice, such as the option of remote working for instance, should be accompanied by a careful listening of the needs of employees that could help tailoring interventions according to their age, to their role, to their marital status. 

## 6. Limitations, Practical Implications, and Conclusions

Despite the significant results discussed above, some limitations need to be addressed to suggest potential future developments for research.

First, the study was cross-sectional and referred to a limited and most specific professional category (employees from the public sector), therefore results cannot not be generalized. Although issues concerning common method variance were controlled both statistically and procedurally, as discussed earlier, a longitudinal design, addressed to follow the same organization across time, would have allowed us to assess the impact of specific HRM interventions on employees’ perceptions and therefore on their organizational behaviors. 

Second, the study adopted self-report measures to collect information on employees’ attitudes and behaviors toward the organization, relying on a partial and subjective view of the variables investigated. Future research could address this limitation by integrating some objective measures of the same constructs (e.g., employees’ participation to training initiatives, supervisors’ assessments of their performance, special leaves requests, working hours reduction, etc.). 

Beside these limitations, as already underlined, the study also revealed some practical implications both for theory and for HRM practices development.

As for the theoretical implications, the study contributed to fostering a positive perspective on the issue of work–life balance, providing empirical support to the growing literature on the enrichment perspective [22] assuming that the involvement in work and in life could be an occasion for individuals to develop skills, to gain knowledge, and to get a sense of fulfilment that could contribute to the general wellbeing of individuals [128,129,130].

As for managerial implications, the study suggested the opportunity to carefully consider work–family interface as a factor that might positively impact on organizational behaviors. HRM practices’ development should take into account employees’ needs in terms of work–life balance (e.g., through dedicated surveys and training initiatives) and consequently target programs (for instance, diversity training aimed at creating an inclusive and unbiased climate, overcoming a traditional gender-based view of the division of labor) that are useful to enhance individual wellbeing, or job crafting interventions aimed to support employees in strategically managing job demands and resources to maximize the person/organization fit, finally impacting on performance. Managerial training could also be useful to support supervisors in developing and sustaining a positive work–family spillover, contributing to creating a family-friendly culture. These HR measures could help organizations to strengthen a work–family organizational culture, based on the perception of supportive and people-based HRM practices [131] that were proved to be significant correlates of several positive organizational behaviors, OCB being among them [80,86,132,133,134,135]. 

## Figures and Tables

**Figure 1 behavsci-12-00301-f001:**
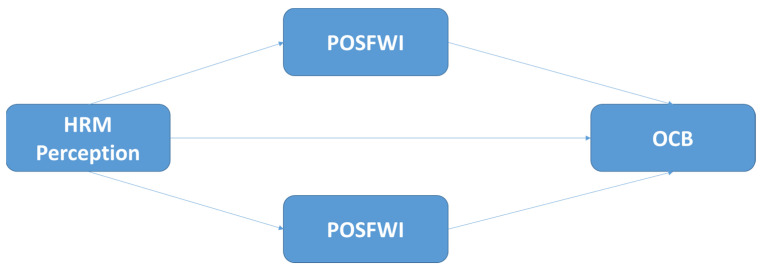
Theoretical mediational model examining the direct and indirect effects of HRM Perception on OCB, mediated by Positive Work-To-Family and Family-To-Work Spillover.

**Figure 2 behavsci-12-00301-f002:**
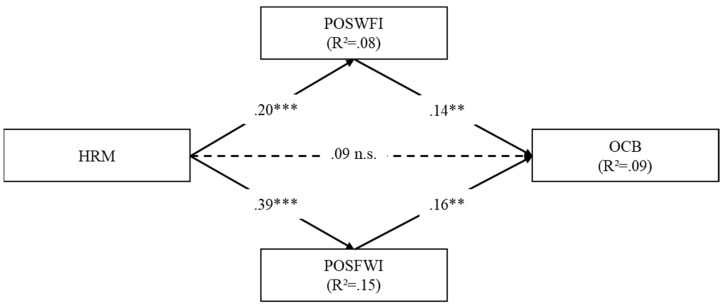
The mediation model. HRM = Human Resource Management. POSWIF = Positive work-to-family spillover. POSFWI = Positive family-to-work spillover. OCB = Organizational Citizenship Behaviours. ** *p* < 0.01. *** *p* < 0.001. The effects of control variables on the mediators and outcome are estimated but not shown for figure clarity.

**Table 1 behavsci-12-00301-t001:** Descriptive statistics and correlations (* *p* < 0.05. ** *p* < 0.01).

Variables	Mean (SD)	1	2	3	4
1. HRM Perception	2.48 (0.62)	(α = 0.84)			
2. POSWFI	2.82 (0.87)	0.18 **	(α = 0.84)		
3. POSFWI	2.51 (0.77)	0.38 **	0.46 **	(α = 0.60)	
4. OCB	4.95 (0.40)	0.18 **	0.25 **	0.26 **	(α = 0.62)

## Data Availability

The data presented in this study are available on request from the corresponding author. The data are not publicly available due to privacy reasons.

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
