# Peer review of "Employees’ Perception of HRM Practices and Organizational Citizenship Behaviour: The Mediating Role of the Work–Family Interface"

_behavsci, 2022, doi:10.3390/bs12090301_

Round 1

Reviewer 1 Report

The paper is scientifically correct and well-structured. For me the weaknesses of the paper are:

- the lack of definition of some concepts: these are not clear to me. The concept of work-family balance is well introduced in the introduction of the text. Other concepts are used without introduction: High Performance Human Resources Management Practices, Organizational Citizenship Behavior, social exchange relationship, green human resources management, perceived service quality.

In the discussion the author links explicitly HRM with people-oriented while this is the core of HRM.  

- the lack of connection to practice: The paper is now too much an exercise in quantitative methodology and trying to find correlations. I want to invite the author to translate the results of this exercise to the workfloor: what does it mean for employees? Why is it a new insight? How can it contribute to the improvement of HRM?  

- the limited explanation of the methodology: the participants all worked for a public administration. It seems important to give more explanation about the participating sector : administration. What are the characteristics of HRM in this administration? What are the problems of work/family balance in this administration (in a public administration the work-family balance is normally not an issue)  

Author Response

Response to Reviewer 1 Comments

Thanks a lot for your precious comments and suggestions that certainly gave us the opportunity to improve our paper and make It more readable.

The paper is scientifically correct and well-structured. For me the weaknesses of the paper are:

- the lack of definition of some concepts: these are not clear to me. The concept of work-family balance is well introduced in the introduction of the text. Other concepts are used without introduction: High Performance Human Resources Management Practices, Organizational Citizenship Behavior, social exchange relationship, green human resources management, perceived service quality.

We added definitions of the unclear concepts

In the discussion the author links explicitly HRM with people-oriented while this is the core of HRM.  

- the lack of connection to practice: The paper is now too much an exercise in quantitative methodology and trying to find correlations. I want to invite the author to translate the results of this exercise to the workfloor: what does it mean for employees? Why is it a new insight? How can it contribute to the improvement of HRM?  

We have clarified the practical implications of the paper

- the limited explanation of the methodology: the participants all worked for a public administration. It seems important to give more explanation about the participating sector : administration. What are the characteristics of HRM in this administration? What are the problems of work/family balance in this administration (in a public administration the work-family balance is normally not an issue)  

We have integrated information regarding the characteristics of the research context

Reviewer 2 Report

The paper  deals with very interesting issues related to work balance policies and organisational aspects. 

The paper is well constructed but I suggest to impone so e aspects:

- in the first part clarify that the compensation model is strictkly related to the Enrichment approach;

- if you speak about public sectpr and the sample is not so large you could clarify with area of public sector you choose, the way you involved the employees and some characteristics of their work positions;

- clarify bitter the elements that characterise the OBC;

- in the conclusions you could suggest some elements to improve the HR measures to improve employees’ well-being.

Author Response

Response to Reviewer 2 Comments

Thanks a lot for your precious comments and suggestions that certainly gave us the opportunity to improve our paper and make It more readable.

The paper deals with very interesting issues related to work balance policies and organisational aspects. 

The paper is well constructed but I suggest to impone so e aspects:

- in the first part clarify that the compensation model is strictkly related to the Enrichment approach;

We clarified the relationship between the compensation model and the Enrichment approach

- if you speak about public sectpr and the sample is not so large you could clarify with area of public sector you choose, the way you involved the employees and some characteristics of their work positions;

We have integrated information regarding the characteristics of the research context and the employees in our possession

- clarify bitter the elements that characterise the OBC;

We described with more detailes the elements of OCB

- in the conclusions you could suggest some elements to improve the HR measures to improve employees’ well-being.

We integrated as suggested

Reviewer 3 Report

The reviewed article deals with the issue of the Employees’ Perception of HRM practices and Organizational Citizenship Behaviour, in particular the the mediating role of Work-Family Interface. The layout of the work is clear, coherent and logical. The division of content and systematics adopted by the author does not raise objections. The research questions were formulated in a precise manner.  Outlined research theses have been well proven. They were correctly verified and presented in a clear way in the conclusions. Research methods were also correctly applied.  The article contains references to the most important publications concerning the title issue.

Author Response

Thanks a lot for your words of appreciation for our research paper.